# Ultrasound Patterns of Hepatocellular Carcinoma and Their Prognostic Impact: A Retrospective Study

**DOI:** 10.3390/cancers15225396

**Published:** 2023-11-13

**Authors:** Chiara Barteselli, Stefano Mazza, Valentina Ravetta, Francesca Torello Viera, Letizia Veronese, Chiara Frigerio, Giulia Gori, Gaetano Bergamaschi, Carmelo Sgarlata, Antonio Facciorusso, Marcello Maestri, Antonio Di Sabatino, Andrea Anderloni

**Affiliations:** 1Gastroenterology and Endoscopy Unit, Fondazione IRCCS Policlinico San Matteo, 27100 Pavia, Italy; 2Department of Internal Medicine and Medical Therapeutics, University of Pavia, 27100 Pavia, Italy; 3First Department of Internal Medicine, Fondazione IRCCS Policlinico San Matteo, 27100 Pavia, Italy; 4Gastroenterology Unit, Department of Medical and Surgical Sciences, University of Foggia, Viale Luigi Pinto 1, 71122 Foggia, Italy; 5General Surgery I, Fondazione IRCCS Policlinico San Matteo, 27100 Pavia, Italy; marcello.maestri@unipv.it

**Keywords:** hepatocellular carcinoma, abdominal ultrasound, ultrasound pattern, overall survival, recurrence-free survival, histological grade

## Abstract

**Simple Summary:**

Abdominal ultrasound (US) is a widely used, first-level technique for the diagnosis of hepatocellular carcinoma (HCC). However, the prognostic meaning of different US patterns of HCC has not been investigated yet. In this study, we present a novel four-type US classification based on HCC morphology; we aimed to assess the impact of US patterns on overall survival (OS) and recurrence-free survival (RFS) in a cohort of HCC patients treated with radiofrequency thermal ablation and with a long-term follow-up. We demonstrated that specific US patterns are independent predictors of both OS and RFS and, therefore, could help identify patients at a higher risk of worse prognosis in clinical practice.

**Abstract:**

Background: Hepatocellular carcinoma (HCC) is a leading cause of cancer-related death. Abdominal ultrasound (US) is by far the most widely used first-level exam for the diagnosis of HCC. We aimed to assess whether different ultrasound patterns were related to tumor prognosis. Methods: We retrospectively reviewed all patients with a new diagnosis of HCC (single nodule) and undergoing radiofrequency thermal ablation (RFTA) at our clinic between January 2009 and December 2021. Patients were classified according to four HCC ultrasound patterns: 1A, single capsulated nodule; 1B, well capsulated intra-node nodule; 1C, cluster consisting of capsulated nodules; and 2, non-capsulated nodule. Results: 149 patients were analysed; median follow-up time was 43 months. US patterns 1A (32.9%) and 1B (61.1%) were the most commonly seen. Median overall survival (OS) and recurrence-free survival (RFS) from RFTA were 54 months (95% CI, 42–66) and 22 months (95% CI, 12–32), respectively. Pattern 1A showed the best OS. Compared to pattern 1A, 1B was independently associated with worse OS (51 months (95% CI, 34–68) vs. 46 months (95% CI, 18–62)) and RFS (34 months (95% CI, 27–41) vs. 18 months (95% CI, 12–24)). Patterns 1C and 2 were associated with worse RFS compared to 1A, while no difference was seen for OS. Among baseline clinical variables, pattern 1B exhibited higher histological grade (*p* = 0.048) and tumor dimension (*p* = 0.034) compared to pattern 1A. Conclusions: Our findings demonstrate that different US patterns correlate with different survival outcomes and tumor behavior in patients with HCC. Prospective studies are needed to confirm these results.

## 1. Introduction

Hepatocellular carcinoma (HCC) accounts for more than 80% of primary liver tumors worldwide [1,2], making it a relevant public health issue. HCC is the sixth most commonly diagnosed cancer and the third leading cause of cancer-related death globally [2,3]. It is among the top three causes of cancer-related deaths in 46 countries and among the top five in 90 countries [2]. Most patients with HCC have an underlying liver cirrhosis, making HCC the leading cause of death in cirrhotic patients. The main risk factors are hepatitis B virus (HBV) and hepatitis C virus (HCV) infections, alcohol abuse causing alcoholic steatohepatitis, and non-alcoholic fatty liver disease (NAFLD) or non-alcoholic steatohepatitis (NASH) [4]. NAFLD represents a rapidly growing HCC etiology due to the obesity epidemic, particularly in western countries, where NAFLD is currently a leading cause of HCC, being associated with about 15–20% of cases [5].

Despite improvements in therapeutic approaches, the five-year and ten-year survival rates are about 20% and 10%, respectively [6]. To define an accurate prognostic system is crucial to guide the therapeutic approach. The Barcelona Clinic Liver Cancer (BCLC) algorithm is the most widely used staging system, with the treatment assigned according to the tumor stage [7]. The BCLC classification is based on HCC radiological features (i.e., nodules number and dimension) along with an assessment of liver function and patient physical fitness [8].

Transabdominal ultrasound (US) is by far the most common imaging technique used to screen and diagnose HCC, thanks to the absence of ionizing radiation and its widespread distribution [9]. An annual HCC incidence of 1–4% in patients with cirrhosis has been reported [10]; thus, biannual screenings are highly recommended [11,12]. In patients with cirrhosis, US proved a sensitivity of 60–80% and a specificity greater than 90% for the diagnosis of HCC [13]. At US examination, HCC nodules may appear more hypoechoic, isoechoic, or hyperechoic than the surrounding parenchyma and may or may not be surrounded by a hypoechoic halo [14]. Besides morphological features, the nodule vascularization can be assessed by contrast-enhanced ultrasound (CEUS), which can reveal a pathognomonic behavior (hyper-enhancing in the early arterial phase, with mild wash-out) [15,16] without the need for histological confirmation [17].

However, to our knowledge, few studies have investigated the predictive role of US imaging features for HCC prognosis; rather, studies have mainly focused on the dynamic and quantitative parameters assessed by CEUS [18,19,20]. To date, no data are available on the correlation between HCC morphological features at US and tumor prognosis.

Thus, the aim of this study was to assess the correlation between the US morphological patterns and survival outcomes in a cohort of patients with HCC.

## 2. Materials and Methods

### 2.1. Study Design and Patients

We performed a single-center, retrospective, observational study. From January 2009 to December 2021, all patients diagnosed with a single nodule of HCC that were treated with radiofrequency thermal ablation (RFTA) were consecutively enrolled. At enrolment, the following data were collected: date of birth, gender, date of diagnosis, etiology of liver disease, Child-Pugh score at diagnosis, degree of HCC on histological examination, size of the nodule, date and cause of death if the patient was deceased, and date of the last follow-up for surviving patients, evaluated up to 1 August 2022 through telephone calls and the informatic system of the Lombardy Region. For final inclusion in the study analyses, patients with at least one of the following criteria were excluded: previous diagnosis of HCC, multifocal HCC, neoplastic thrombosis, metastatic lymph nodes, ASA score > 3, or mixed HCC-cholangiocarcinoma (CCC) histotypes according to a biopsy examination. Patients lost to follow-up, whose survival could therefore not be calculated, were also excluded. We included only patients treated with the same therapeutic approach (i.e., RFTA) in order to obtain a homogeneous cohort, thereby avoiding biases related to different therapies. The follow-up time was calculated between the date of RFTA and the date of death or last follow-up. A minimum follow-up time of 6 months was considered.

The study was approved by the local Ethics Committee and was performed according to the Helsinki declaration. All patients gave their written informed consent for inclusion in the study, clinical data collection, and for the conservation of biologic materials.

### 2.2. US Patterns Definition

The US morphological subtypes of HCC nodules were developed before starting the enrollment, based on the following experience. Over the years, all the ultrasound-detected HCC nodules that were candidates for surgical treatment were evaluated by intraoperative ultrasound before their surgical removal. We tried to correlate the ultrasound appearance of the nodule with its macroscopic appearance when removed and dissected. Through this experience, four ultrasound patterns were individuated. We defined as 1 the category of capsulated nodules, divided as follows: 1A—single, capsulated nodule; 1B—intra-node node, well capsulated; and 1C—cluster formation consisting of capsulated nodules. All nodules without a capsule, i.e., with ill-defined margins, were included in category 2. Example images of the four US nodule subtypes with the relevant macroscopic appearance after surgical resection are shown in Figure 1. Patients were assigned to one of these groups based on a double-blind evaluation of the US images, performed by two experienced sonographers (more than 300 US performed annually). In case of discrepancies, the case was revised by third skilled operator, and a final decision was reached after a collegial discussion.

### 2.3. Study Objectives

The primary aim of the study was to investigate a correlation between ultrasound patterns and overall survival. The secondary aims were to correlate the ultrasound pattern with recurrence-free survival and with histological grade and other baseline clinical variables.

### 2.4. Statistical Analysis

The categorical variables were absolute frequency and percentage. The continuous variables with normal distribution were mean ± SD, whereas the continuous variables without normal distribution were median and range. Time-to-event data, namely, overall survival (OS) and recurrence-free survival (RFS), were calculated from the first procedure until the event (i.e., death or recurrence) or the last follow-up visit and were estimated by means of Kaplan-Meier analysis. The correlation between US patterns and survival outcomes was assessed by means of Cox regression analysis; the results were expressed as hazard ratio (HR) and 95% CI. Specifically, survival analyses were performed by calculating the hazard ratio of death; the US pattern exhibiting better survival in the one-by-one Kaplan-Meier analysis was taken as the reference for comparisons with other patterns (i.e., two-by-two comparisons). Variables found to be significant in the univariate analysis were entered in the multivariate model. Correlations between the US patterns and the baseline clinical variables were assessed using the Mann-Whitney analysis and the Chi-Square or Fisher’s exact tests for continuous and categorical variables, respectively. Analyses were performed using R Statistical Software “Survival” package (Foundation for Statistical Computing, Vienna, Austria), and significance was established at the 0.05 level (two-sided).

## 3. Results

### 3.1. Baseline Features

Between January 2009 and December 2021, 149 patients received a diagnosis of HCC and were treated with RFTA (median age at enrollment 73 ± 8 years, 59.7% male). The predominant HCC etiology was HCV (76.5%). The most represented ultrasound pattern was 1B (61.1%). Only 3.3% and 2.7% of patients demonstrated patterns 1C and 2, respectively. All of the demographic and baseline clinical features of the patients are reported in Table 1. The median follow-up time was 43 ± 33 months.

### 3.2. Survival Analyses

During follow-up, calculated between the date of RFTA and the date of death or last follow-up, 102 patients (68.5%) died, of whom 64.7% died due to liver-related disease (liver failure or progression of oncological disease). The 1- and 5-year cumulative probabilities of OS from RFTA were 91% (95% CI, 86-96) and 48% (95% CI, 40-56), respectively, with a median OS of 54 months (95% CI, 42-66) (Figure 2).

One hundred and two patients (68.5%) experienced an HCC recurrence during follow-up, with a median RFS of 22 months (95% CI, 12-32) and 1-year and 5-year cumulative probabilities of RFS of 66% (95%CI, 58-74) and 27% (95%CI, 19-35), respectively.

### 3.3. Correlation between US Patterns and Survival Outcomes

The median OS for the different US pattern were as follows: 62 months (95%CI, 41–83) for pattern 1A, 51 (95%CI, 34–68) for pattern 1B, 46 (95%CI, 18–62) for pattern 1C, and 40 (95%CI, 29–53) for pattern 2 (Figure 3a). The 1A patter was therefore set as the reference for the two-by-two comparison with other US patterns. In our univariate regression analysis (Table 2), Child-Pugh class B (HR 1.97; 95%CI, 1.19–3.27; *p* = 0.008) and C (HR 15.17; 95%CI, 3.38–68.04; *p* < 0.001), nodule size (HR 1.02; 95%CI 1.01–1.03; *p* = 0.010), and ultrasound pattern 1B (HR 2.74; 95%CI, 1.32–5.69; *p* = 0.006) were found to be correlated with increased mortality. In our multivariate regression analysis (Table 2), all those variables proved to be independent risk factors for mortality: HR 1.99; 95%CI, 1.18–3.37; *p* = 0.009 for Child-Pugh class B; HR 22.07; 95%CI, 4.69–103.92; *p* < 0.001 for Child-Pugh class C; HR 1.02; 95%CI, 1.00–1.03; *p* = 0.020 for nodule size; and HR 2.50; 95%CI, 1.18–5.30; *p* = 0.010 for US pattern 1B.

The median RFS for US pattern 1A was 34 months (95%CI, 27–41), 18 months for 1B (95%CI, 12–24), 9 months for 1C (95%CI, 3–14), and 11 months for 2 (95%CI, 0–26) (Figure 3b). The univariate and multivariate regression analyses (Table 3) showed that Child-Pugh class B and C (HR 2.99; 95%CI, 1.32–4.37; *p* = 0.001 and HR 4.11; 95%CI, 2.69–10.87; *p* < 0.001, respectively), nodule size (HR 1.32; 95%CI, 1.40-3.13; *p* = 0.040), and all ultrasound patterns (HR 2.54; 95%CI 1.10–4.30; *p* = 0.02 for 1B; HR 3.24; 95%CI, 1.43–7.93; *p* = 0.01 for 1C; HR 2.29; 95%CI, 1.09–6.80; *p* = 0.020 for 2) were independently associated with increased mortality.

### 3.4. Correlation between US Patterns and Baseline Clinical Variables

The four US patterns were homogeneous for all the baseline variables recorded (Table 4). Patterns 1A and 1B were then compared separately. By dichotomizing the histological grade between G1 and. G2+G3, we found that pattern 1B was correlated with a higher histological grade compared to pattern 1A (Table 5). Moreover, patients with pattern 1B were characterized by higher median tumor dimension compared to patients with pattern 1A (22 mm vs 25 mm, respectively; *p* = 0.034) (Figure 4). All other variables showed no differences between the two US patterns.

## 4. Discussion

This is the first study to investigate the prognostic value of HCC morphology as assessed by transabdominal US. Additionally, we present a novel US classification of HCC nodules based on the two main groups of capsulated and non-capsulated nodules, with the capsulated ones being further divided into three subgroups. These patterns had been defined in the years before this study and correspond not only to the US image but also to the true macroscopic morphology of the tumor when dissected after surgical resection; this may therefore make our US classification and our results more reproducible. It is also worth noting that the four ultrasound patterns were homogeneous for all the baseline parameters recorded and that we considered only patients treated with the same therapeutic approach (i.e., RFTA), thereby reducing the risk of biases. We evaluated relevant outcomes, including the overall survival and the recurrence-free survival.

Patterns 1A (single, capsulated nodule) and 1B (well capsulated, intra-node nodule) were by far the most commonly found, covering roughly 95% of the cases. In our survival analysis, pattern 1A showed the highest survival probabilities over time and was used as a reference to compare with other patterns. As the main finding, our study showed that patterns 1B, 1C (nodule cluster appearance), and 2 (non-capsulated nodule) were independently correlated with worse prognosis. Particularly, pattern 1B was associated with increased overall mortality, while all three patterns correlated with increased risk of recurrence over time. Patterns 1C and 2 failed to demonstrate a correlation with OS, possibly because of the small sample size of these groups. The only analogy we could find in the literature was with a previous study assessing the prognostic role of preoperative US features in a cohort of HCC patient undergoing liver resection, in which the authors found that a mixed echogenicity pattern was associated with shorter survival [21]. This similarity is of interest, since in our classification, pattern 1B and pattern 1C are clearly characterized by inhomogeneous, mixed echo patterns compared with pattern 1A.

Other studies have evaluated the prognostic impact of CEUS quantitative parameters on survival outcomes, showing that some variables, such as rise time and time to peak, were associated with worse OS and RFS [18,19]. However, the application of these parameters in clinical practice is limited, since they cannot be interpreted in real time but require post-examination analysis.

In agreement with existing data in the literature [22,23], our univariate and multivariate analysis confirmed that nodule size and Child-Pugh class are independent risk factors for OS and RFS. In our study, Child-Pugh class C played the most important role in determining patient prognosis, with the highest HR both for OS and RFS. It must be specified that, as recommended by the BCLC model [8,24], we applied RFTA only for patients with preserved liver function. Only two patients underwent RFTA despite being in Child-Pugh class C, but they were characterized by a good performance status and belonged to the ultrasound pattern 1A group, i.e., with encapsulated nodules that were easily distinguishable from the surrounding liver parenchyma.

Regarding histological grade, although the guidelines recommend biopsy only when a nodule > 10 mm has a doubtful contrast enhancement behavior in two imaging methods in cirrhotic patients [7], in our sample, biopsies were performed in most patients. This was due to the two HCC study protocols that were applied in our center during the study period, i.e., requiring histological confirmation, and to the fact that we usually perform a biopsy at the first diagnosis of a liver nodule to exclude cholangiocarcinoma or mixed HCC- cholangiocarcinoma histotypes, for which different therapeutic strategies should be undertaken [25,26]. As stated, no correlations were found between the US patterns and the baseline variables; however, we dichotomized the tumor grading between G1 and G2+G3 and found that pattern 1B was correlated with a higher HCC histological grade compared to pattern 1A. We failed to demonstrate a similar result for patterns 1C and 2. Additionally, HCC grade was not correlated with OS in our study, in contrast to data previously reported in literature [27]. We would explain this discrepancy with the small sample size, particularly for the G3 group.

Finally, our results confirm the prognostic role of tumor size, which was proved to be independently correlated with both OS and RFS. This finding agrees with several previous reports [28,29,30,31], although other studies did not seem to confirm such an association [32,33]. Additionally, we found that pattern 1B had significantly higher tumor size compared to pattern 1A, supporting a possible difference in tumor behavior among the different US patterns.

This study had some limitations: its retrospective nature, which makes it necessary to confirm our results in larger, prospective cohorts; the small sample size of US patterns 1C and 2, which, although sufficient to demonstrate a correlation with RFS, limited our ability to analyze such a correlation with OS. The strengths of the study were the homogeneity of the study cohort, consisting of patients all treated with the same HCC therapeutic approach and regularly followed up thereafter, the relevance of the analyzed outcomes, and the long-term median follow-up that made these outcomes more robust.

## 5. Conclusions

In conclusion, our study demonstrates, for the first time, that the US pattern of HCC nodules at diagnosis may be interpreted as an indicator of OS, RFS, and histological grade. Further studies are needed to confirm these data prospectively in order to assess if the US pattern can help in identifying a subgroup of patients at increased risk of worse outcome, thus aiding the selection of a suitable therapeutic approach and follow-up.

## Figures and Tables

**Figure 1 cancers-15-05396-f001:**
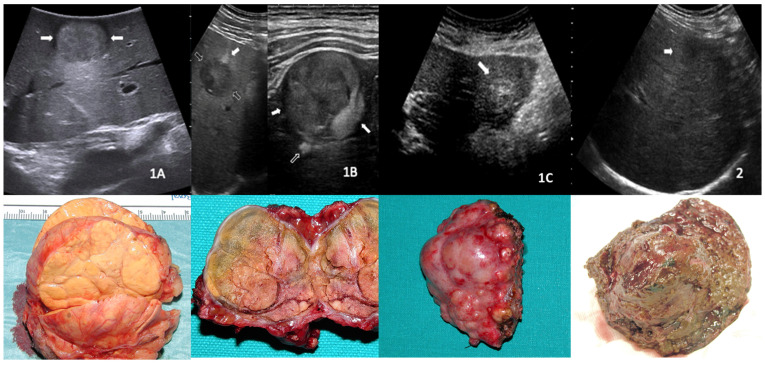
Illustrative images of hepatocellular carcinoma nodules divided into the four ultrasound patterns, with the relevant macroscopic appearance showed below. (**1A**)—single, capsulated nodule; (**1B**)—intra-node node, well capsulated; (**1C**)—cluster formation consisting of capsulated nodules; and (**2**)—non-capsulated.

**Figure 2 cancers-15-05396-f002:**
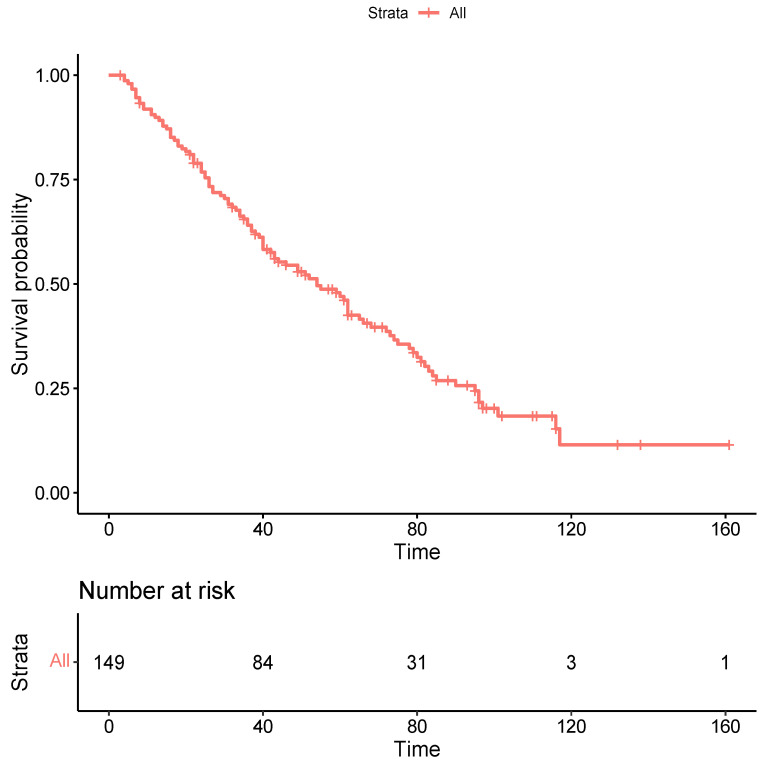
Kaplan-Mejer curve for overall survival. Time expressed in months.

**Figure 3 cancers-15-05396-f003:**
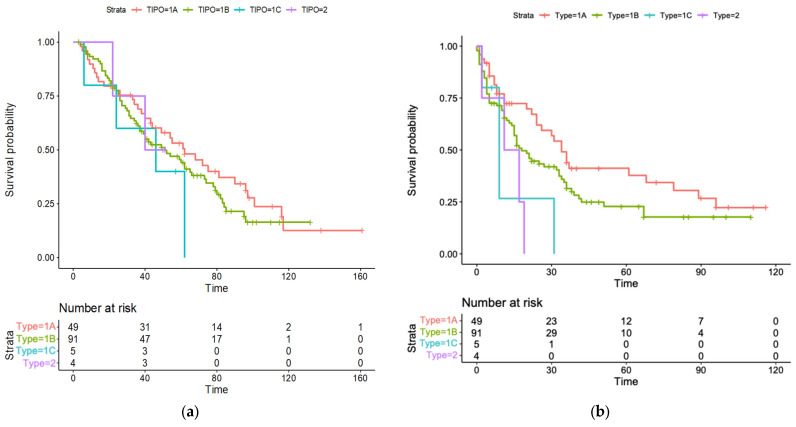
Kaplan-Mejer curves for overall survival and recurrence-free survival stratified by ultrasound patterns. (**a**) Overall survival; (**b**) Recurrence-free survival.

**Figure 4 cancers-15-05396-f004:**
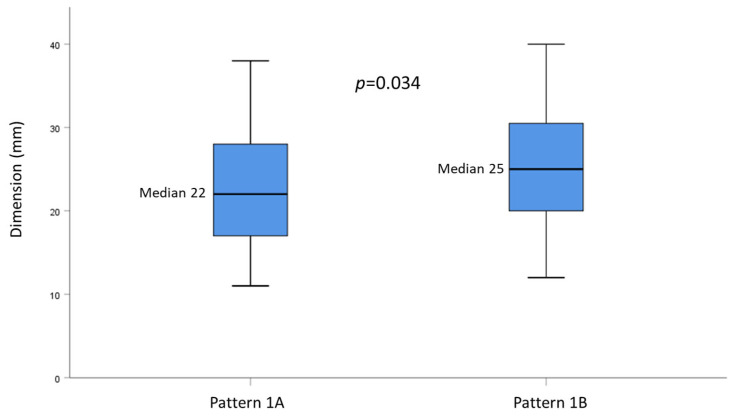
Box plot graph showing the distribution of nodule sizes according to US pattern 1A or 1B.

**Table 1 cancers-15-05396-t001:** Demographic and baseline clinical characteristics of the 149 patients analyzed.

Parameter	N = 149
Age, years, mean (SD)	73 (7.6)
Sex, n (%) Male Female	89 (59.7) 60 (40.3)
Aetiology, n (%) Alcohol HBV HCV NASH Other	15 (10.1) 9 (6.0) 114 (76.5) 7 (4.7) 4 (2.7)
Child-Pugh class, n (%) A B C	124 (83.2) 23 (15.4) 2 (1.4)
Histological grade, n (%) No biopsy G1 G2 G3	17 (11.4) 46 (30.9) 75 (50.3) 11 (7.4)
Nodule size, mm, median (range)	24 (11–68)
Ultrasound pattern, n (%)	
1A	49 (32.9)
1B	91 (61.1)
1C	5 (3.3)
2	4 (2.7)

NASH, nonalcoholic steatohepatitis; HBV, hepatitis B virus; HCV, hepatitis C virus; SD, standard deviation.

**Table 2 cancers-15-05396-t002:** Univariate and multivariate analysis of baseline predictors of overall survival.

Features	Univariate Analysis		Multivariate Analysis	
	HR (95% CI)	*p* Value ^1^	HR (95% CI)	*p* Value ^1^
Age	1.01 (0.99–1.04)	0.174		
Sex (reference female)	1.29 (0.87–1.92)	0.193		
Aetiology (reference HBV) Alcohol HCV NASH Other	1.21 (0.78–1.54) 0.88 (0.45–1.74) 1.32 (0.36–4.85) 0.69 (0.19–2.51)	0.677 0.725 0.671 0.589		
Histological grade (reference G1) G2 G3	1.27 (0.80–2.00) 1.64 (0.77–3.48)	0.321 0.209		
Child-Pugh class (reference A) B C	1.97 (1.19–3.27) 15.17 (3.38–68.04)	0.008 ^2^ <0.001 ^2^	1.99 (1.18–3.37) 22.07 (4.69–103.92)	0.009 ^2^ <0.001 ^2^
Nodule size	1.02 (1.01–1.03)	0.010 ^2^	1.02 (1.00–1.03)	0.020 ^2^
Ultrasound pattern (reference 1A) 1B 1C 2	2.74 (1.32–5.69) 2.11 (0.73–6.14) 1.27 (0.29–5.38)	0.006 ^2^ 0.169 0.743	2.5 (1.18–5.30)	0.010 ^2^

^1^ Cox regression analysis; ^2^ Statistically significant; CI, confidence interval; NASH, nonalcoholic steatohepatitis; HBV, hepatitis B virus; HCV, hepatitis C virus; HR, hazard ratio; SD, standard deviation.

**Table 3 cancers-15-05396-t003:** Univariate and multivariate analysis of baseline predictors of recurrence-free survival.

Features	Univariate Analysis		Multivariate Analysis	
	HR (95% CI)	*p* Value ^1^	HR (95% CI)	*p* Value ^1^
Age	1.04 (0.96–1.08)	0.213		
Sex (reference female)	1.33 (0.79–1.83)	0.328		
Aetiology (reference HBV) Alcohol HCV NASH Other	0.90 (0.45–1.79 0.89 (0.42–1.78) 0.78 (0.36–3.20) 1.03 (0.45–2.20)	0.764 0.754 0.651 0.595		
Histological grade (reference no biopsy) G1 G2 G3	0.98 (0.42-1.71) 1.32 (0.68–2.19) 2.11 (0.87–3.87)	0.676 0.347 0.110		
Child-Pugh class (reference A) B C	2.91 (1.34–4.11) 3.17 (1.38–8.15)	0.006 ^2^ 0.002 ^2^	2.99 (1.32–4.37) 4.11 (2.69–10.87)	0.001 ^2^ <0.001 ^2^
Nodule size	1.23 (1.13–2.02)	0.010 ^2^	1.32 (1.40–3.13)	0.040 ^2^
Ultrasound pattern (reference 1A) 1B 1C 2	2.30 (1.20–4.60) 3.11 (1.73–5.70) 2.27 (1.15–4.82)	0.020 ^2^ 0.008 ^2^ 0.010 ^2^	2.54 (1.10–4.30) 3.24 (1.43–7.93) 2.29 (1.09–6.80)	0.020 ^2^ 0.010 ^2^ 0.020 ^2^

^1^ Cox regression analysis; ^2^ Statistically significant; CI, confidence interval; NASH, nonalcoholic steatohepatitis; HBV, hepatitis B virus; HCV, hepatitis C virus; HR, hazard ratio; SD, standard deviation.

**Table 4 cancers-15-05396-t004:** Characteristics of the 149 patients stratified by ultrasound pattern.

Ultrasound Pattern	1A	1B	1C	2	*p* Value ^1^
Number	49	91	5	4	
Age, years, mean (SD)	73.94 (8.27)	72.46 (7.30)	78.60 (4.16)	77.25 (4.92)	0.173
Sex, n (%)					0.819
Male	27 (55.1)	56 (61.5)	3 (60)	3 (75)	
Female	22 (44.9)	35 (38.5)	2 (40)	1 (25)	
Aetiology, n (%)					0.647
Alcohol	6 (12.2)	8 (8.8)	1 (20.0)	0 (0.0)	
HBV	3 (6.1)	6 (6.6)	0 (0.0)	0 (0.0)	
HCV	38 (77.6)	69 (75.8)	4 (80.0)	3 (75.0)	
NASH	0 (0.0)	6 (6.6)	0 (0.0)	1 (25.0)	
Other	2 (4.1)	2 (2.2)	0 (0.0)	0 (0.0)	
Child-Pugh class, n (%)					0.337
A	40 (81.6)	78 (85.7)	3 (60.0)	3 (75.0)	
B	7 (14.3)	13 (14.3)	2 (40.0)	1 (25.0)	
C	2 (4.1)	0 (0.0)	0 (0.0)	0 (0.0)	
Histological grade, n (%)					0.509
No biopsy	7 (14.3)	10 (11.0)	0 (0.0)	0 (0.0)	
G1	20 (40.8)	24 (26.4)	1 (20.0)	1 (25.0)	
G2	18 (36.7)	51 (56.0)	3 (60.0)	3 (75.0)	
G3	4 (8.2)	6 (6.6)	1 (20.0)	0 (0.0)	
Nodule size, mm, median (range)	22 (11–50)	25 (12–68)	30 (22–45)	24 (16–36)	0.150

^1^ Chi-Square or Fisher’s exact tests for categorical variables, Mann-Whitney analysis for continuous variables; CI, confidence interval; NASH, nonalcoholic steatohepatitis; HBV, hepatitis B virus; HCV, hepatitis C virus; HR, hazard ratio; SD, standard deviation.

**Table 5 cancers-15-05396-t005:** Correlation between ultrasound pattern (1A and 1B) and histological grade.

	1A	1B	*p*-Value
Histological grade, n (%)			**0.048**
G1	20 (47.6%)	24 (29.6%)	
G2/G3	22 (52.4%)	57 (70.4%)	

## Data Availability

The research data are stored in an institutional repository and will be shared upon request to the corresponding author.

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
