# Peer review of "Ultrasound Patterns of Hepatocellular Carcinoma and Their Prognostic Impact: A Retrospective Study"

_cancers, 2023, doi:10.3390/cancers15225396_

Round 1
Reviewer 1 Report
Comments and Suggestions for Authors
The manuscript entitled " Ultrasound Patterns of Hepatocellular Carcinoma and Their Prognostic Impact: a retrospective study. " has been reviewed.
This is a very interesting paper.
A few questions.
Hepatocellular carcinoma and cholangiocarcinoma are embryologically different, but can they be treated in the same way?
Is Child-Pugh an appropriate parameter for validation with echographic findings?
Comments on the Quality of English Language
none
Author Response
The manuscript entitled " Ultrasound Patterns of Hepatocellular Carcinoma and Their Prognostic Impact: a retrospective study. " has been reviewed.
This is a very interesting paper.
A few questions.
Hepatocellular carcinoma and cholangiocarcinoma are embryologically different, but can they be treated in the same way?
Thank you for the comment. To our knowledge, despite hepatocellular carcinoma (HCC) and cholangiocarcinoma (CCC) share some therapeutic approaches, mainly liver resection, and despite an overlap in the anatomical region, the two diseases are really different in terms of staging and prognosis. Moreover, locoregional treatments (such as radiofrequency ablation (RFTA)) and liver transplantation, that are a curative option for HCC, are not an option for CCC. Thus, we excluded patients with a mixed HCC-CCC pattern at histology, since these patients are characterized by a worse prognosis and different therapeutic approach. Additionally, we enrolled only HCC patients treated with RFTA to reduce the bias that could have resulted from the effect of different treatments on survival outcomes.
Is Child-Pugh an appropriate parameter for validation with echographic findings?
Thanks a lot for the observation. We are sorry we didn’t make it clear. Actually, the reference for the US pattern in the univariate and multivariate analyses was the pattern 1A, which was the US pattern showing the best survival. We have made it clearer both in the “Statistical analysis” paragraph and in the Results section.
Reviewer 2 Report
Comments and Suggestions for Authors
Barteselli et al., in the present manuscript investigated the prognostic value of hepatocellular carcinoma (HCC) morphology by trans-abdominal ultrasound (US). The authors presented a new US classification of HCC nodules based on capsulated and non-capsulated tumor features. The authors further divided the capsulated group into three main categories. The patterns of tumors were identified based on US image and macroscopic morphology dissected after surgical resection, thus making the US classification and the analysis potentially more accurate.
The manuscript is well written and organized and it has a clinical value. The big limitation is the small size of the study groups.
It would be useful if the authors may include information about the consent obtained from the patients.
The manuscript would benefit of representative images of “true macroscopic morphology” of the tumor after surgical resection and US images.
Do the authors envision to apply deep learning to their system?
Comments on the Quality of English Language
Minor English editing is required.
Author Response
Barteselli et al., in the present manuscript investigated the prognostic value of hepatocellular carcinoma (HCC) morphology by trans-abdominal ultrasound (US). The authors presented a new US classification of HCC nodules based on capsulated and non-capsulated tumor features. The authors further divided the capsulated group into three main categories. The patterns of tumors were identified based on US image and macroscopic morphology dissected after surgical resection, thus making the US classification and the analysis potentially more accurate.
The manuscript is well written and organized and it has a clinical value. The big limitation is the small size of the study groups.
Thank you for the comment. We agree about this limitation, particularly regarding the small sample size of the 1C and 2 pattern groups, which is related to the fact that most patients with HCC have nodules falling into the 1A and 1B pattern groups. Despite a correlation with the OS didn’t emerge for the 1C and 2 patterns possibly due to a non-adequate study power, we were however able to demonstrate a correlation with the RFS, which is a relevant outcome in these patients.
It would be useful if the authors may include information about the consent obtained from the patients.
Thank you for the comment. All patients signed an informed consent for the collection of clinical data and for the conservation of biologic materials. A statement about the informed consent has been added in the “Materials and methods” section.
The manuscript would benefit of representative images of “true macroscopic morphology” of the tumor after surgical resection and US images.
Thank you very much for the important comment, which gave us the opportunity to improve the Figure 1. We have retrieved the representative images of the macroscopic appearance of HCC nodules after surgical resection, one for each US pattern, and we have integrated them into the Figure 1.
Do the authors envision to apply deep learning to their system?
This is a very interesting point and we thank you for the comment. Deep learning is a rapidly evolving field in medicine and is contributing to the progress in image recognition. We think that deep learning will be increasingly present in our clinical practice in the next years, and more studies are desirable to have more robust data. To our knowledge, deep-learning-based methods, particularly the convolutional neural network, have recently been applied for the diagnosis of liver parenchyma disease (such as steatosis) and for the differentiation of liver lesions (e.g., HCC versus other liver tumor types) in several studies. Our results first need validation in prospective studies, but as a perspective it would be very interesting to apply deep learning to our system, for the differentiation of HCC nodules subtypes.
Reviewer 3 Report
Comments and Suggestions for Authors
In the present study titled “Ultrasound Patterns of Hepatocellular Carcinoma and Their Prognostic Impact: a retrospective study” authored by Barteselli et al., the authors demonstrated that certain ultrasound (US) patterns serve as independent predictors of both overall survival (OS) and recurrence-free survival (RFS). These findings suggest that these patterns can aid in identifying patients at a higher risk of experiencing a worse prognosis in clinical practice. The abstract is well-written and conclusive. However, there are a few minor corrections that need to be addressed.
First, the authors should consider incorporating recent worldwide data on hepatocellular carcinoma (HCC) cases into their study.
Second, what is TIPO in Figure 3a?
Additionally, there appears to be a typographical error in Table 2, where "183other" is mentioned. This may need correction.
What is reference number 33? Recheck in the reference list.
Author Response
In the present study titled “Ultrasound Patterns of Hepatocellular Carcinoma and Their Prognostic Impact: a retrospective study” authored by Barteselli et al., the authors demonstrated that certain ultrasound (US) patterns serve as independent predictors of both overall survival (OS) and recurrence-free survival (RFS). These findings suggest that these patterns can aid in identifying patients at a higher risk of experiencing a worse prognosis in clinical practice. The abstract is well-written and conclusive. However, there are a few minor corrections that need to be addressed.
First, the authors should consider incorporating recent worldwide data on hepatocellular carcinoma (HCC) cases into their study.
Thank you for the comment. We have updated the worldwide data on HCC according to the most recent evidence available in the literature. In fact we have added in the “Introduction” a relevant reference published in the current year about the global epidemiology of HCC (Singal, A.G et al. Global trends in hepatocellular carcinoma epidemiology: implications for screening, prevention and therapy. Nat Rev Clin Oncol. 2023).
Second, what is TIPO in Figure 3a?
Additionally, there appears to be a typographical error in Table 2, where "183other" is mentioned. This may need correction.
Thanks a lot for the observations. The Figure 3a and the Table 2 have been corrected in the revised manuscript.
What is reference number 33? Recheck in the reference list.
Thank you. We have rechecked the reference list and the citations in the text and we confirmed that the total reference number of the submitted manuscript was 32. With the above-mentioned added reference now the total reference number is 33.
Round 2
Reviewer 2 Report
Comments and Suggestions for Authors
The authors addressed all the concerns and the manuscript is fine.